# Effects of microplastic and microglass particles on soil microbial community structure in an arable soil (Chernozem)

Katja Wiedner[1] and Steven Polifka[2]

[1] SEnSol - Sustainable Environmental Solutions Consulting UG, Gleichen, Germany

[2] Physical Geography and Land Use Systems, Department of Geography, Ludwig-Maximilians-University, Munich, Germany

*Correspondence to:* Katja Wiedner (kawi.science@googlemail.com)

**Abstract**

Microplastic and microglass particles from different sources enter aquatic and terrestrial environments. The complexity of its environmental impact is difficult to capture and consequences on ecosystem components e.g. soil microorganisms are virtually unknown. Addressing this issue, we performed an incubation experiment by adding 1% of five different types of impurities ($\leq 100\ \mu m$) to an agricultural used soil (Chernozem) simulating a worst-case scenario of contamination. The impurities are made of polypropylene (PP), low density polyethylene (LD-PE), polystyrene (PS), polyamide12 (PA12) and microglass. After 80 days of incubation at 20°C, we examined soil microbial community structure by using phospholipid fatty acids (PLFA) as markers for bacteria, fungi and protozoa. The results showed that soil microorganisms were not significantly affected by the presence of microplastic and microglass. However, PLFAs tend to increase in LD-PE (28%), PP (19%) and microglass (11%) treated soil in comparison with untreated soil, whereas PLFAs in PA12 (32%) and PS (11%) treated soil decreased. Interestingly, PLFAs revealed significant differences PA12 (-89%) and PS (-43%) in comparison to LD-PE. Furthermore, variability of bacterial PLFAs was much higher after microplastic incubation whereby fungi seemed to be unaffected from different impurities after 80 days of incubation. Similar results were shown for protozoa, which were also more or less unaffected by microplastic treatment indicated by minor reduction of PLFA contents compared to control. In contrast, microglass seems to have an inhibiting effect on protozoa because PLFAs were under the limit of determination. Our study indicated, that high amounts of different microplastics may have contrary effects on soil microbiology. Microglass might have a toxic effect for protozoa.

## 1. Introduction

Microplastics are used e.g. for a range of consumer products or industrial application such as abrasives, filler, film and binding agents. The identification and quantification of sources and pathways of microplastics into the environment are highly diverse and difficult to detect. While different methods have been developed for synthetic polymer identification and quantification in sediments and water, analytical methods for soil matrices are still lacking or in an early experimental stage (e.g. Hurley et al., 2018). It is assumed that microplastics enter (agricultural) soils with soil amendments, irrigation and the use of agricultural plastic films for mulch applications, but also through flooding, atmospheric deposition and littering (Bläsing and Amelung, 2018; Hurley and Nizzetto, 2018; Kyrikou and Briassoulis, 2007; Ng et al., 2018; Weithmann et al., 2018). The extent of microplastics polluted soil ecosystems is probably much higher than previously thought. For instance, a recent study by Weithmann et al. (2018) found 895 plastic particles (> 1 mm) per kilogram dry weight in digestate from a biowaste digester used as soil fertilizer after aerobic composting. Li et al. (2018) detected an average microplastic concentration of 22.7 ± 12.1 $^{x}$ $10^3$ particles per kilogram dry weight in 79 sewage sludge samples from 28 wastewater treatment plants in China. The total amount of microplastics already entered soil habitats is uncertain, but Ng et al. (2018) estimated that 2.3 to 63.0 Mg ha$^{-1}$ microplastic loadings from biosolids reached agroecosystems.

Properties of microplastics differ regarding its size, morphology, origin and chemical composition. A generally accepted definition for the term "microplastics" does not exist so far although essential for industry, research and political decision-makers. In several studies, microplastics are only defined as particles < 5 mm (5000 μm) and a contradistinction to nanoparticles is seldom given in environmental studies. Some environmental studies, however, specify microplastics in large (1 mm to 5 mm) and small (1 μm to 1 mm) particles (Wagner et al., 2014). The term "nanoplastic" and its definition is still controversial discussed. Gigault et al., (2018) specified nanoplastics and recommend 1 μm as upper size limit.

Microplastic particles are differentiated into primary microplastics (e.g. for abrasives, cosmetic additives or industrial resin pellets) and degraded secondary microplastics, which result from formerly larger plastic debris. Microplastic particles could be highly diverse regarding its morphology leading to a varying effects in environmental systems (Wagner et al., 2014).

More than 200 different types of plastic are known, which may have different properties e.g. regarding its reactivity or bioavailability in soil environment. Thus, differentiation of microplastic should not only base on size but also regarding its chemical (e.g. hydrophobicity scales) and physical properties (e.g. morphology) may affecting physicochemical soil properties and soil biology. For instance, De Souza Machado et al. (2018) showed, that 2% microplastic concentration in soil affects bulk density, water holding capacity, hydraulic conductivity, soil aggregation, water stable aggregates and microbial activity. This comprehensive study elucidates the complexity of processes triggered by the presence of microplastic particles in soil environment. Microglass is currently not part of the microplastics discussion although glass is very resistant to corrosion or weathering and can be thought as corrosion-proof (Papadopoulos and Drosou, 2012). Microglass is used as blasting abrasive, filling material and an additive of road markings. Thus, it enters the environment on similar ways than microplastics e.g. in sewage sludge or abrasive from roads. The effects on terrestrial ecosystems are equally unknown as those of microplastics. The difficulty of highly diverse study structures and test environments due to heterogenic material properties is already reported in related research disciplines like marine and freshwater ecology (Phuong et al., 2016; Rist and Hartmann, 2018). To create a standardize study structure in soil science, we highly recommend for future scientific studies dealing with the effect of artificial microparticles on soil flora and fauna to use the definition and size.

Furthermore, a detailed description of microparticle characteristics should be mandatory to show potential
interactions between biotic or abiotic soil components and microparticles on different size scales.
The present study contributes to a deeper understanding of the impact of different types of microplastics and
microglass (~100 μm) on soil microbial community structure in an agricultural soil. For this, different types of
microplastics and microglass were added to arable soil and incubated for 80 days. In order to identify possible
shifts in the microbial community structure we used phospholipid fatty analysis (PLFA). This study was guided
by the following research questions:

1. Is it possible to observe distinct shifts in microbial community due to the presence of microparticles?
2. Do different plastic material properties stimulate microbial groups in diverse ways?
3. Does microglass affect the microbial community in a similar way to microplastics?
**2. Material and Methods**
**2.1 Soil sampling and incubation experiment**
Soil samples were taken on March 11, 2018 near Brachwitz (51°31'46" N, 11°52'41" E; 102 m above sea level),
10 km northwest of Halle (Saale) (Saxony-Anhalt, Germany). The samples were randomly taken at four different
spots (A, B, C, D) from the first 10 cm of an arable topsoil in order to have four independent replicates, which
served as basic substrate for the incubation experiment. Soil was immediately sieved (< 2 mm) after sampling and
divided into subsamples for further basic soil analytics. Subsample material used for incubation was stored at
approximately 8°C. The soil subsamples were set at a water content of 60% water holding capacity (WHC) and
pre-incubated for three weeks at 20°C.
A respective amount of 1% (w/w) of polypropylene (PP), low density polyethylene (LD-PE), polystyrene (PS),
polyamide12 (PA12) (Rompan, Remda-Teichel, Germany) and microglass (Kraemer Pigmente GmbH & Co.KG,
Aichstetten, Germany) was added to each independent soil replicate and stirred manually for homogenization with
a glass stirring rod. This quantity is equal to 12.6 Mg microparticles ha$^{-1}$ (bulk density topsoil: 1.26 g cm$^{-3}$)
indicating worst-case scenario. However, a study by Fuller and Gautam (2016) found similar contaminated soils
closed to industrial areas. In addition, a control soil replicates were incubated without additives of microplastics
or microglass. Due to the use of arable topsoil as incubation substrate, a microplastic contamination cannot be
excluded. However, due to the high microplastic loads used in this the experiment a possible prior contamination
is negligible. Microplastics were not pre-treated to cause degradation (e.g. with ultraviolet radiation) to simulate
primary microplastic particles in soils. Incubation was performed in laboratory bottles for 80 days at 20°C without
daylight. During this period all bottles were opened weekly for 30 s in order to secure aerobic conditions.
Furthermore, the total weight of each bottle was monitored. In the case of any weight loss, an equivalent amount
of water was replenished to provide a constant water holding capacity of 60%. According to manufacturer
specifications sizes of microplastic and microglass particles ranged between 90-100 μm. The microplastics used
in this study are commonly used for daily products and cosmetics (bottle caps, drinking straws (PP), plastic bags,
milk bottles, food packaging film (LD-PE), disposable cups, packaging materials (PS), inks and clothing (PA))
and were detected in high amounts in sewage sludge of Lower Saxony (Mintenig et al., 2017; Shah et al., 2008).

## 2.2 Soil basic properties

For soil basic characterization, soil subsamples (not samples for incubation) were air dried and sieved (< 2 mm). Total carbon (TC) and total nitrogen (TN) analysis were carried out with a vario Max cube CNS analyzer (Elementar Analysensysteme GmbH, Langenselbold, Germany). Electrical conductivity (EC) and pH values were analyzed by using suspensions of 0.01 M $CaCl_2$ and distilled $H_2O$ at a soil solution ratio of 1 to 2.5. Soil particle size distribution was measured in a suspension using a Helos/KR laser diffractometer (Sympatec GmbH, Clausthal-Zellerfeld, Germany) equipped with a Quixel wet dispersion unit (Sympatec GmbH, Clausthal-Zellerfeld, Germany). Before analysis the sample material was treated with a dispersing agent (0.2 M tetra-Sodium diphosphate decahydrate). For the evaluation of water holding capacity, 10 g of soil was weighted into a plastic cylinder with fine-mesh on the bottom and placed in water. After 24 hours, saturated samples were drained until water release stopped and weighted again for calculation of water holding capacity. Soil subsamples used for determination of soil basic properties were not used for incubation experiment.

Soil chemical properties of the Chernozem topsoil (IUSS Working Group WRB, 2015) were as follows: Total organic carbon (TOC) $28.6 \pm 1.8$ g $kg^{-1}$, Total nitrogen (TN) $2.48 \pm 0.13$ g $kg^{-1}$, C:N $11.56 \pm 0.15$, EC $170 \pm 9$ µS $cm^{-1}$ and $pH_{CaCl2}$ $5.13 \pm 0.02$. Proportions of clay, silt and sand were $7.0 \pm 0.2$ %, $58.5 \pm 3.6$ % and $34.5 \pm 3.7$ %, respectively and the soil texture was classified as silt loam (FAO, 2006). Water holding capacity was $0.218 \pm 0.005$ $g_{H2O}$ $g_{dry\ weight}^{-1}$.

## 2.3 Phospholipid fatty acid analysis

Phospholipid fatty acid (PLFA) analyses were performed using a modified version of the Bligh and Dyer method (Frostegård et al., 1993). 6 g of fresh soil were extracted with a single-phase trichloromethane/methanol/citrate buffer system (1:2:0.8; v/v/v). 19:0 was added as first internal standard (IS1) to each sample for later quantification of the phospholipids. Extracts were centrifuged for 15 minutes at 4000 rpm. The supernatants were separated using a liquid-liquid extraction. Lipid fractionation was performed using a silica based solid phase extraction. Remaining phospholipid fractions of the samples and the external standards were treated by an alkaline saponification using 0.5 M sodium hydroxide in methanol followed by a methylation with boron trifluoride in methanol (12%). A liquid-liquid extraction with saturated sodium chloride solution and hexane was used to separate the organic phase, which contains the fatty acid methyl esters. For quality control 5-α-cholestane was added as second internal standard (IS2) after the phase separation. Analytes were transferred with isooctane into GC autosampler vials and analyzed by a GC 2010 capillary gas chromatograph (Shimadzu Ltd., Tokyo, Japan) equipped with Supelco SPB-5 fused silica capillary column (30m x 0.25 mm x 0.25 µm film thickness) and flame ionization detector. All PLFA contents were corrected for dry mass due to the use of fresh soil for extraction. For this purpose, WHC was determined subsequent to sample weighing.

Single PLFA were assigned to taxonomic groups according to following pattern: Total fungi: 18:2ω6,9, 18:1ω9c; protozoa: 20:4ω6c; general bacteria: 14:0, 15:0, 16:0, 17:0, 18:0; gram-positive bacteria: i14:0, a14:0, i15:0, a15:0, i16:0, a16:0, i17:0, a17:0; gram-negative bacteria: 16:1ω7c, cy17:0, 18:1ω7c, cy19:0; Actinomycetes (ACT): 10Me18:0 (Frostegård et al., 1993; Olsson et al., 1999; Zelles, 1999; Zelles et al., 1992). These biomarkers are not entirely specific for their taxonomic groups and therefore must be interpreted cautiously (Zelles, 1997). For total bacteria the sum of general, gram-positive, gram-negative and ACT was calculated. Sum of PLFA describes the sum of measured contents of fungal-derived, bacterial-derived, protozoa and the unspecific PLFA markers 16:1ω5c and 10Me16:0.

148

**2.4 Scanning Electron Microscopy (SEM)**

Microplastic samples were fixed on an object slide and coated with gold using a Q150R ES rotary pumped sputter coater (Quorum Technologies Ltd., Laughton, United Kingdom) in a low vacuum atmosphere. The SEM images were taken with a Tabletop Microscope TM4000Plus (Hitachi Ldt., Tokyo, Japan).

**2.5 Statistical analysis**

Statistical analysis and graphical design were carried out using R 3.5.0 (R Core Team, 2018). Prior test assumption of normally distributed data was examined using Shapiro-Wilk test. Because of mostly non-normal distributed data Brown-Forsythe test was used for checking for homoscedasticity in the groups. Residuals of each linear model were checked graphically for homoscedasticity and normal distribution to validate the model performance. Because of widespread heteroscedasticity and bad model performances, differences in PLFA marker contents between treatments of each taxonomic microbial group were statistically evaluated using the Kruskal-Wallis rank sum test. Dunn's test was performed for multiple comparison between the treatment levels in case of a significant ($p \leq 0.05$) treatment effect in the Kruskal-Wallis test (Dunn, 1964). Holm method was used to control the family-wise-error rate caused by the pairwise multiple comparisons (Holm, 1979). Different lowercase letters were used to illustrate significant differences between homogeneous subsets. Interquartile range of boxplot whiskers is 1.5.

# 3. Results

## 3.1 Morphology and size of microparticles

The SEM images of the microplastics (PP, LD-PE, PS, PA12) and microglass are shown in Fig. 1, illustrating the heterogenic morphology between but also within the same type of microplastic. Furthermore, according to the manufacturer specifications size of microplastics and microglass should range between 90 to 100 μm. Many particles are, however, much bigger (up to 200 μm) or smaller (down to 10 μm). Especially LD-PE, PA12 and PP have a slag-like structure leading to pore formation, whereas PS has a plate shaped structure with fringed or even sharp edges. Pointy and sharp edges are also shown for LD-PE, PA12 and PP. In contrast, microglass particles appear with a few exceptions more regular than the microplastic ones and could be described as microspheres.

## 3.2 Impact of microplastics and microglass on soil microbial community structure

The total PLFA contents do not show significant differences between single specific microparticles compared to the control (Fig. 2). Nevertheless, the PLFA contents of microglass, PP and LD-PE treated soil tend to increase compared to the control by 11, 19, and 28%, respectively, whereas PA12 and PS show lower PLFA contents compared to the control by 32 and 11%. The comparisons of single plastic types show that PLFA contents of PA12 and PS are with 89% and 43%, respectively, significant lower compared to LD-PE (Fig. 2). A similar pattern is also observable in treatment distribution of each group PLFA content of bacteria and fungi. Although, the fungi show a more inexplicit pattern compared to bacteria. This might imply that positive and negative stimulations of the single microplastics affect bacteria as well as fungi in a similar way. Compared to the control bacterial-derived PLFA contents show an increase in soil treated with microglass (19%), PP (25%) and LD-PE (32%). On the other hand, a decline of total bacteria has been determined in soil treated with PA12 (-33%) and PS (-11%, Fig. 3). Fungal PLFA contents, however, show a smaller increase compared to the control by 9% (microglass), 15% (PP),

24% (LD-PE) and a lower decrease by -22% (PA12) and -9% (PS; Fig. 3). The treatment effect variability of
bacterial-derived PLFAs are multiple times higher compared to fungal-derived PLFAs. For instance, the highest
positive median deviation of total bacterial-derived PLFAs to the control is 32% (LD-PE), whereas the highest
negative deviation is 33% (PA12). In contrast, positive deviation of fungal-derived PLFAs compared to the control
is only 24% (LD-PE) and negative deviation is only 22% (PA12, Fig. 3).
Regarding a whole comparison of all treatments, with the exception of protozoa, the increase of PLFA contents
could be observed for all fungal and bacterial (Gram-negative, Gram-positive, ACT, general) groups when
incubated with microglass, LD-PE and PS (Fig. 3). The significant lower PLFA contents of PA12 compared to
LD-PE are also shown continuously in all microbial groups (Fig. 3). In contrast to the fairly consistent pattern of
the fungi and bacteria, protozoa show a different pattern. Protozoa PLFA contents decreased for all microplastics
by up to 21% (LD-PE) compared to the control (Fig. 3). PA12 and PP show a comparatively high data variability
compared to the other treatments. Most interestingly, PLFA content of protozoa was under the limit of
determination for all replications incubated with microglass.

## 4. Discussion

High amounts of artificial soil impurities (12.6 Mg microplastics or -glass ha$^{-1}$) do not have a significant effect on
soil microbial community structure within the incubation time of 80 days. However, there is a conspicuous
tendency that different types of microplastics may have promoting (LD-PE, PP) or reducing effects (PA12, PS) on
soil microorganisms (Fig. 2 and 3). Furthermore, different plastics have obviously various effects on individual
taxonomic groups as indicated by the significant lower values of treatment PA12 and PS compared to LD-PE (Fig.
2 and 3). As mentioned in Section 3.2, the variability of bacterial-derived PLFA are much higher than fungal-
derived PLFAs, which possibly indicates that bacteria are more susceptible to interference. However, this is not
surprisingly because bacteria respond relatively fast on environmental changes (e.g. changing water conditions,
temperature, etc.) e.g. due to their rapid reproduction rate (e.g. Fierer et al., 2003).
Reasons for missing significant effects between microparticle treatments and the untreated control after 80 days
may be found in the conscious choice of primary microplastics, which were not pre-treated to cause a physical
degradation (e.g. ultraviolet radiation). Subsequently, microplastics are mostly chemically inert during the
experiment due to unaltered chemical and physical properties, which e. g. prohibit the exposition of potential
ecotoxic compounds. Nevertheless, the treatment of soil by different microparticles causes changes in microbial
communities, albeit not significant. The observed effects are based on complex soil-impurity interactions and
studies dealing with the impact of microplastics on soil microbiology are still lacking (Rillig and Bonkowski,
2018; Zhang et al., 2019) and, to our best knowledge, published PLFA or even DNA based studies are still missing.
However, de Souza Machado et al. (2018) investigated the microbial activity after the addition of different amounts
of polyester and polyacrylic fibers as well as polyethylene fragments by measuring the enzyme activity with
fluorescein diacetate (FDA). The study showed that polyester and polyacrylic fibers reducing microbial activity
whereas the soil incubated with polyethylene fragments showed no clear tendency. The effects might be caused
e.g. through changes in soil bulk density, water holding capacity or aggregate changes (de Souza Machado et al.,
2018). The reasons for the observed promoting and also inhibiting effects on microorganisms from different plastic
types, remain a matter of speculation and further research is necessary addressing these issues. The causes
mentioned by de Souza Machado et al. (2018) are essential reasons effecting soil microbiology.
Nevertheless, the morphology and surface properties of microplastics should not be underestimated. The slag-like
structure of LD-PE, PA12 and PP form wrinkles and pores (Fig. 1) and may act as habitat for soil microorganisms.
This in turn may have a promoting effect on the soil microbial community composition of soil as known from pore
rich soil additives e.g. such as charcoal (biochar). For instance, fungal hyphae or bacteria penetrate in pores and
wrinkles and are protected from predators (Lehmann et al., 2011; Thies and Rillig, 2009). Furthermore,
McCormick et al. (2014) showed that microplastic particles could act as habitat for bacteria in rivers.
Umamaheswari et al. (2014) found fungi hyphae from *Penicillium sp., Fusarium sp.* and *Aspergillus sp.*, which
colonized and grew on the surface of soil buried PS after 70 days. The potential colonization of microorganism on
the surface of LD-PE was clearly reviewed by (Kumar Sen and Raut, 2015), who also mentioned the penetration
of the microplastic surface by fungi hyphae. Similar colonization of bacteria were reported by Harrison et al.
(2014), who found rapid attachment of microorganisms onto LD-PE microplastics within coastal marine sediments
after 14 days. In sum, LD-PE seems to benefit the bacterial and fungal colonization. Both bacteria and fungi tend
to increase populations in our experiment. LD-PE may also act as habitat as well as carbon source. The extent of
these functions is mostly controlled by abiotic for example ultraviolet irradiation and temperature (Kumar Sen and
Raut, 2015). Thus, the provided habitat seems to be the most important factor for enhanced PLFA in our
experiment, because abiotic factors were either excluded (no ultraviolet irradiation) or kept constant (stabile
temperature at 20°C). However, colonization on microplastic surfaces after incubation was not determined in this
experiment and currently it is still uncertain, if colonized microplastic surface areas could also act as a hotbed for
extensive soil colonization. Furthermore, it remains uncertain why PA12 seems to inhibit microorganisms in this
experiment though having similar surface properties as e.g. LD-PE, which tends to promote the microorganisms.
According to Galloway et al. (2017), organic compounds, nutrients and pollutants can accumulate on microplastic
surface in aquatic ecosystems. It can be assumed that this also occurs in terrestrial ecosystems such as soil
environments. Furthermore, it is conceivable that also humic substances accumulate on microplastic surfaces
leading to an increased colonization of specific microorganisms and in consequence to the formation of a bacterial
biofilm. The accumulation of nutrients and water on a surface is the precondition for the formation of biofilms
consisting of extracellular polymeric substances derived from bacteria (Flemming and Wingender, 2010). The
formation of biofilms may occur within three weeks, as shown by Lobelle and Cunliffe (2011) investigated the
surface of PE particles in marine environment. Due to the constant (water)conditions in this study, the formation
of biofilms on microplastic surfaces cannot be excluded at least on LD-PE and PP particles as well as microglass
indicating promoting effects on soil microorganisms reflected by increased PLFA contents. Future research on the
role of artificial microparticles in soil microcosm is urgently needed to clarify potential risks, intensities of soil
microbiological disturbance by microplastics due to promoting colonization of specialized (and harmful)
microorganism, toxicity due to released harmful chemicals or a direct damage after entering microorganism as
secondary nanoparticles (Lu et al., 2019).
Beside the morphology of microplastic, its surface chemistry has effects on soil physicochemical processes. In
comparison to LD-PE, PP and PS, which show hydrophobic characteristics, PA12 combines hydrophobic and
hydrophilic surface groups (Schmidt et al., 2015) whereby microglass has a hydrophilic surface. A study by
Marangoni et al. (2018) showed, that glass microspheres (4 µm, 7-10 µm and 30-50 µm; micoglass addition of 1-
5% v/v) reduced the mobility of water reflected in a large decrease of the spin-spin relaxation time of water protons,
decreases in the self-diffusion coefficient of water molecules, a lower water activity, and strengthening of O-H
bonds. The study further showed that glass microspheres have an inhibiting effect on *Escherichia coli* growth and

the germination of *Medicago sativa* seeds. In our experiment, an inhibiting effect of microglass could not be shown for the most microorganisms with the exception of protozoa (Fig. 3). Based on the results by Marangoni et al. (2018) is conceivable that protozoa respond in a similar way to the presence of microglass like *Escherichia coli*. Nevertheless, these harmful effects of microglass particles on protozoa observed in our study are surprisingly, because this indicates that e.g. sand grains in soil, which consist of $SiO_2$, may also have inhibitory effects on protozoa. To our best knowledge no studies were performed in order to investigate this question.

Another important fact is the heterogeneity of microplastics. The wide variance between the several types of plastic and just as the heterogeneity of different sources prevent a generalization of scientific results. For example Cao et al. (2017) visualized polystyrene using SEM. The showed image of PS differs strongly from the plastic used in this study. The way of producing, the pathway to environment and the degradation status of microplastics play an important role for evaluating the behavior of microplastics in soil or other environments. Furthermore, it remains ambiguous if primary microplastics added to soils cause similar effects compared to secondary microplastics, which result from the decomposition of larger plastic debris. Depending on the parent plastic material and environmental variables, highly diverse plastic surfaces result from uncontrolled surface modification due to decomposition processes. This fact is already known from the comparison of primary and secondary nanoplastics properties (Gigault et al., 2018). Especially in view of the fact that already emitted macro- and microplastics will degrade in terrestrial ecosystems right up to nanoscales.

Nevertheless, it should be borne in mind that PLFA analyses and laboratory experiments always generate limited results. Fast change of PLFA pattern only allows a determination of actual state of the microbial community structure and it is unreliable to use single PLFA biomarker for taxa detection, which is feasible by deoxyribonucleic acid (DNA) analyses. But compared to gene sequencing or other DNA analyses, PFLA biomarker analysis is rapider and cheaper (Frostegård et al., 2011). Another problem may be the transferability of results generated on laboratory scale under ideal conditions (well-known homogenous plastic fabrics as treatments, simplified and controllable regimes, no rhizosphere, etc.). Also, the single addition of high amounts of microplastics does not reflect the ordinary way how microplastics enter an ecosystem. The accumulation of plastic particles in soils is rather a long and gradual process than a single event, which do not trigger sudden environmental impacts (Rillig et al., 2019). Thus, this first study should only serve as a basic work, which stimulates future microbial studies dealing with microparticles in soils or sediments. So, further research is needed to link laboratory and environmental conditions to enhance the environmental relevance of microplastic research. High amounts were chosen to show worst-case effects on highly contaminated place (industrial areas or floodplains in vicinity of urban areas). On the other hand, agricultural land is treated regularly with compost, sewage sludge and other microplastics containing soil amendments or plastic mulches are used in vegetable production. Due to their recalcitrance plastic tend to accumulate in soil. So, a worst-case scenario is able to illustrate future soil statuses on an undefined time scale.

## 5. Conclusion

This study aimed the question, whether high amounts of microplastics and microglass have effects on soil microbial community structure by using PLFAs as microbial markers. High amounts were added to soil in order to show a worst-case scenario in highly contaminated soils (e.g. industrial areas or floodplains in vicinity of urban areas). On the other hand, agricultural land is treated regularly with compost, sewage sludge and other

microplastics containing soil amendments. Furthermore, plastic mulches used for fruit and vegetable production are further sources of microplastic in soils. Due to its high recalcitrance, plastic tend to accumulate in soil. Thus, our worst-case scenario may illustrate future soil statuses at an undefined time scale. The use of microbial markers in laboratory incubation experiments, describing microbial soil communities always act as a simplification of complex natural environmental systems. This study provides first insights into soil microcosm disturbed by different microparticles. The results provide hints that after 80 days of incubation microorganisms are either promoted or inhibited depending on the type of the impurities. Different microplastic types seem to have contrary effects on soil microorganisms depending on the origin and the properties of the plastics, which influence the morphological and chemical appearance of the microplastics. On the other hand, microglass seems to be even highly toxic for protozoa. Within this study we cannot clarify why bacteria and protozoa show different reaction on quartz glass microparticles. Changes in soil microbiology induced by plastic pollution have unexpected consequences for soil ecosystems. This study should therefore be considered as basis for further research which is urgently needed in order to understand the long-term consequences of microplastics in soils and other terrestrial ecosystems.

**Data availability.**

All data compiled in this study is published in figures. Detailed primary data and underlying research are available by request from the corresponding author.

**Author contributions.**

KW conceptualized and carried out the experiment. Laboratory work was performed by KW and SP. Statistical analysis and data visualization was carried out by SP. KW prepared the manuscript with contributions from SP.

**Competing interests.**

The authors declare that they have no conflict of interest.

**Acknowledgements.**

We acknowledge Bruno Glaser (Soil Biogeochemistry, Martin Luther University Halle-Wittenberg) for providing the Soil Biogeochemistry laboratories for PLFA analysis. We are grateful to Aline Brosch for supporting the incubation experiment. Furthermore, we thank Tobias Bromm (Soil Biogeochemistry, Martin Luther University Halle-Wittenberg) for supporting the GC measurements. We also thank Gregor Borg (Economic Geology and Petrology, Martin Luther University Halle-Wittenberg) and his scientific staff Andreas Kamradt and Tim Rödel for providing the SEM. Finally, we are grateful to the anonymous reviewers and editor for their critical and detailed comments.

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

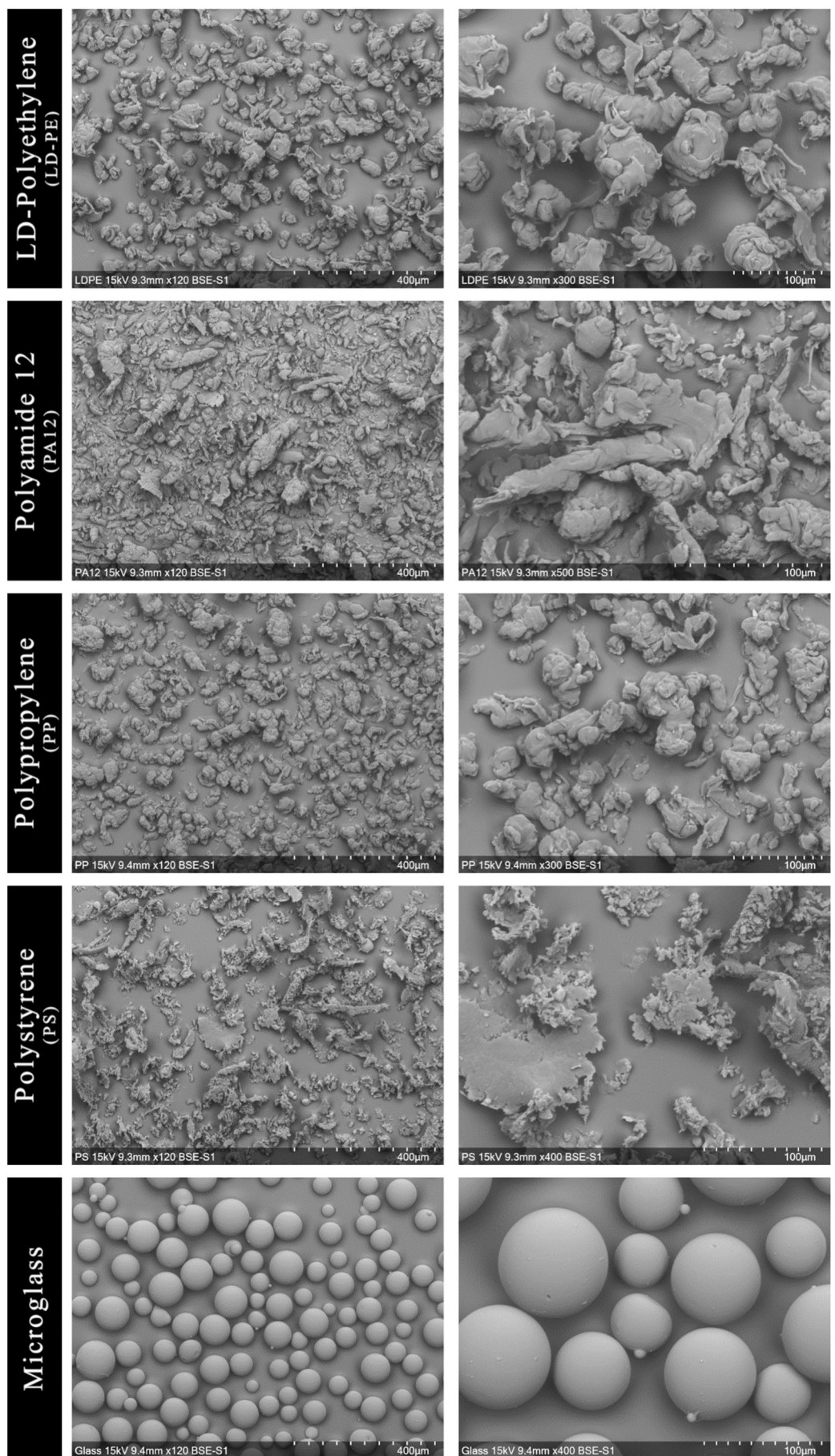

**Figure 1.** Heterogenic particle size distribution and morphology depending on the microparticle type visualized by scanning electron microscopy (SEM).

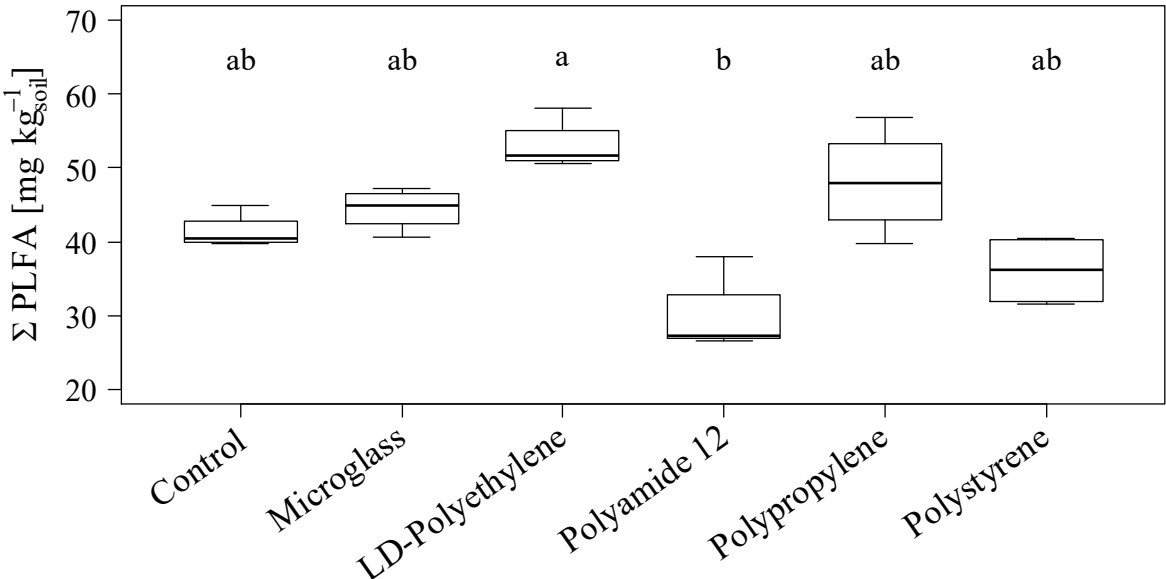


**Figure 2.** Sum of total phospholipid fatty acids as microbial marker in an incubated Chernozem after 80 days.
Different lowercase letters indicate significant differences between the treatments according to a multiple
comparison by Dunn's test (n=4, p < 0.05).

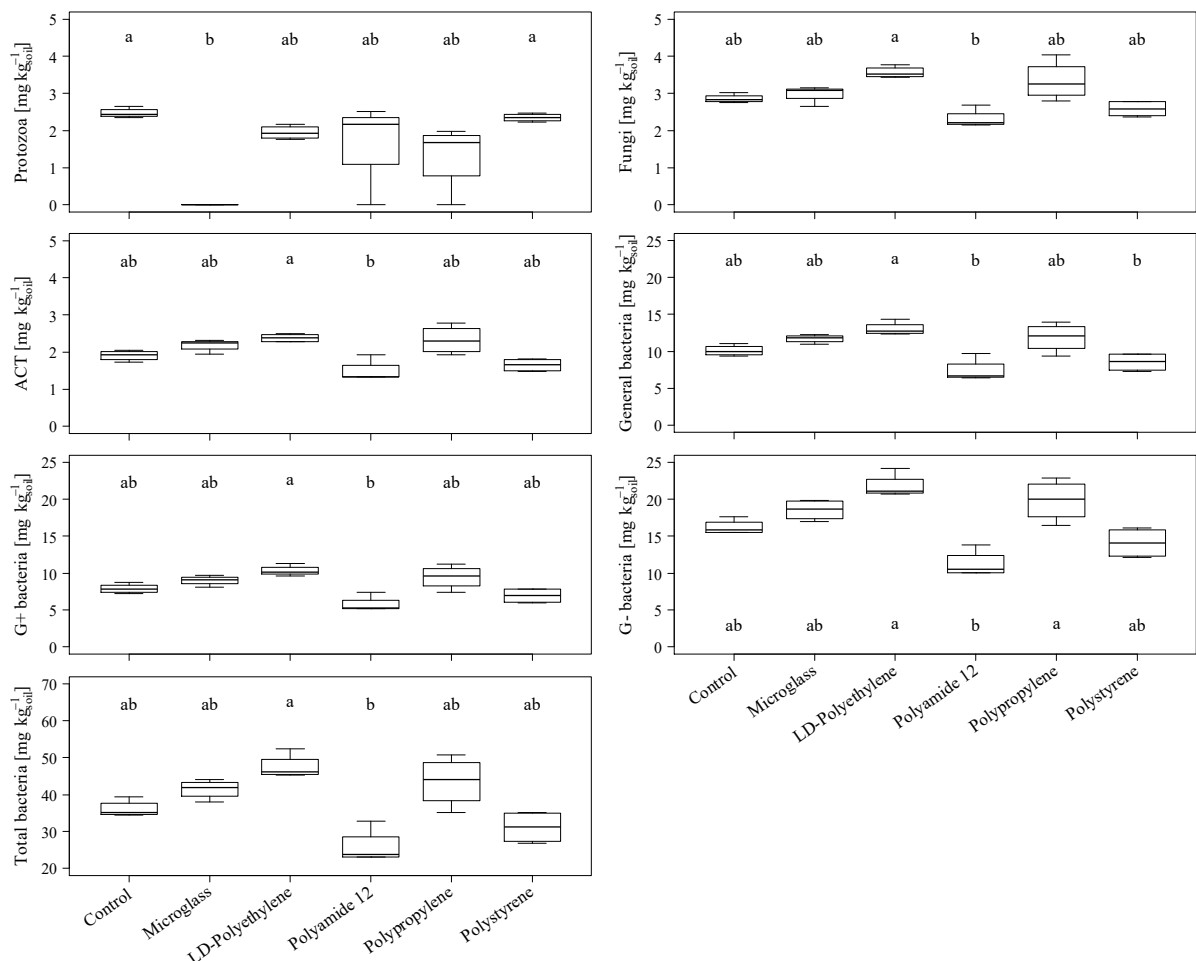


**Figure 3.** Microbial-derived phospholipid fatty acid contents of the individual taxonomic groups of an incubated

Chernozem after 80 days. Different lowercase letters indicate significant differences the treatments according to a
multiple comparison by Dunn's test (n=4, p < 0.05). Please note varying ordinate scales.