# Peer review of "Effects of microplastic and microglass particles on soil"

_SOIL, 2019_

## Referee Comment (RC1) · Anonymous Referee #1 · 16 Jul 2019

Increasing loads of microplastic waste potentially burden our soils. In this regard the paper is timely, as it investigates potential effects of microplastic and microglass pollution on soil microbial community in a laboratory incubation study. The manuscript is concise, very well written and organized, and it has improved in regard to a previous version. However, still the paper includes the risk of presenting artificial results, which should be very openly discussed. The shortcomings refer to: 1. Microplastic loads: The authors state that they refer to microplastic loads near industrial areas. However, 12 t ha-1 is a huge amount, far from being realistic. The authors should spell out clearly, also in abstract and conclusions, that their data refer to worst-case conditions that do not necessarily apply to common plastic and microglass loads in

agricultural soils, because concentrations exceed natural loads at least by a factor of about 10.000! 2. I like the finding for protozoa, and appreciate that an explanation is offered related to the hydrophilic surface. Nevertheless, why should this apply to glass but not to increased amounts of sand grains? Can enhanced amoutns of quartz grains also be toxic for protozoa and has this been published beforeß And if not, why should the glass be more toxic than pure sand? Here the authors should elaborate the physiological explanations a bit more in detail and also outline why microglass should be toxic whereas quartz particles in the fine sand fraction is apparently not (or is it?). It is also not clear why a specific toxicity should only apply for protozoa while one of their main food sources, bacteria, are not affected. 3. Experiment conditions: Usually soil has to be stored cool but should not be air-dried. Air-drying soil prior to incubation in known that it includes the risks of artifacts, even if pre-incubated. The authors should discuss this issue based on some literature which investigated related effects of sieving and air-drying for a range of microbial parameters

Some minor comments: - L 164: Do not show any instead "show no"

- . 204; What do you mean by !"trend" Please, show p-value

- PLFA are only biomarkers, not as sensitive as DNA analyses for specific taxa. The authors should be careful in taking each PLFA biomarker for granted, and they should add a discussion on potential misinterpretations and uncertainties, maybe in an extra paragraph towards the end of the methods section.

- Note that 10Me16:0 is not only used for Actinomycetes, for instance, but has largely been suggested for S utilizing bacteria (see, e.g., work done by R. Evershed and others)

- Figure 1 is nice but it does not really relate to the contents of this paper. If he authors want to leave it, I suggest they should go a bit more into detail into the consequences of comparing the different sizes.

- The stirring for microglass and microplastic incorporation into soil likely interfered with soil aggregation? Can it be that this stirring jointly with glass treatment also impaired protozoa? For me this would be a reasonable explanation for the results presented. . ..

---

## Referee Comment (RC2) · Anonymous Referee #2 · 25 Nov 2019

This manuscript investigates the effects of microplastic and microglass particles on the structure of microbial communities in soil using soil microcosms that have been spiked with these contaminants. The issue of micro-particles in the environment is very topical, and while there is a lot of information about the impact of macro-plastics on wildlife e.g. marine animals, there is relatively little information on the impact of microparticles on microbial populations in terrestrial environments. In this respect, this manuscript is timely. However, the explanation of the experimental design was lacking, and therefore the results should be interpreted with care.

Are the authors confident that an incubation period of 80 days was sufficient to observe

full effects of the addition of micro-particles? How was 80 days selected as the end point of the experiment? Was it based on published literature or observations? The apparent lack of significant alterations in the bacterial and fungal communities may be due to a relatively short incubation time. In addition, the authors did not consider the effects of transfer of the field soil into the lab environment and compartmentalisation of the soils as a cause of the observed changes in PLFAs. This could be remedied if the authors provide PLFA profiles before the soils were used in the microcosms for comparison, or consider such changes in the discussion section.

The amounts of microparticles used in the microcosms (1%) is very high compared to what is observed in the field. The authors state that this is comparable to an industrial site, but this is a rare case, and so these results will not be relevant for most environmental scenarios.

If the authors think that colonisation on the microplastics could explain the increase in PLFAs, they could use SEM to confirm this, especially when they had already used SEM to characterise the micro-particles at the beginning of the experiment.

In the discussion section, the authors discuss the changes of PLFAs after the addition of microparticles, but also state that overall, soil organisms were not significantly affected. If the latter is true, then the relative changes are of no consequence. Instead, there should be a discussion on the apparent lack of impact of microplastics on the microbial communities, especially when the literature that they cite points to the contrary. On the other hand, the PLFA may not be able to detect finer microbial community changes that e.g. a DNA-based method will be able to detect – there needs to be a discussion on this. There should also be more of a discussion on why microglass should only affect protozoa and not bacteria. The authors only cite one paper, but it confuses matters as they found that microglass inhibits bacterial growth, which was not the case in the experiment.

Minor points:

The manuscript could do with a native English speaker to correct the grammar.

The paragraph in the discussion section on the effects of micro-particles on macro-fauna seems irrelevant when the experiments were about testing microbial populations.

Figure 1 does not add anything to the manuscript.

I don't understand the use of lowercase a and b to denote p-values. Better to state the p-values.

The use of the plastic cylinders to adjust water holding capacity will also contaminate the soils with plastic.

'WHC' should be defined.

---

## Author Comment (AC1) · 28 Feb 2020

Response to comments of anonymous referee #1

Referee comments:

Increasing loads of microplastic waste potentially burden our soils. In this regard the paper is timely, as it investigates potential effects of microplastic and microglass pollution on soil microbial community in a laboratory incubation study. The manuscript is concise, very well written and organized, and it has improved in regard to a previous

version. However, still the paper includes the risk of presenting artificial results, which should be very openly discussed.

The shortcomings refer to:

1. Reviewer: Microplastic loads: The authors state that they refer to microplastic loads near industrial areas. However, 12 t ha-1 is a huge amount, far from being realistic. The authors should spell out clearly, also in abstract and conclusions, that their data refer to worst-case conditions that do not necessarily apply to common plastic and microglass loads in Discussion paper agricultural soils, because concentrations exceed natural loads at least by a factor of about 10.000!

Response: We include line 14-15, 306 and 314 (Abstract, Discussion, Conclusion) that the amounts of microparticles used in our study indicating a worst-case scenario.

2. Reviewer: I like the finding for protozoa, and appreciate that an explanation is offered related to the hydrophilic surface. Nevertheless, why should this apply to glass but not to increased amounts of sand grains? Can enhanced amoutns of quartz grains also be toxic for protozoa and has this been published before? And if not, why should the glass be more toxic than pure sand? Here the authors should elaborate the physiological explanations a bit more in detail and also outline why microglass should be toxic whereas quartz particles in the fine sand fraction is apparently not (or is it?). It is also not clear why a specific toxicity should only apply for protozoa while one of their main food sources, bacteria, are not affected.

Response: We conceive the idea of the reviewer and tried to find studies dealing with effects of natural or artificial particles, which are made of quartz, on protozoa. To our best knowledge no studies were performed in order to investigate this question. We added a critical evaluation of this result in line 259-261.

"...This harmful effects of microglass particles on protozoa observed in our study are surprisingly, because this indicates that e.g. sand grains in soil, which consist of SiO2,
may also have inhibitory effects on protozoa. To our best knowledge no studies were performed in order to investigate this question. ..."

3. Reviewer: Experiment conditions: Usually soil has to be stored cool but should not be air-dried. Air-drying soil prior to incubation in known that it includes the risks of artifacts, even if pre-incubated. The authors should discuss this issue based on some literature which investigated related effects of sieving and air-drying for a range of microbial parameters

Response: The reviewer is right. However, the soil samples in our experiment were not air-dried prior to incubation. In our opinion, the reviewer misunderstood the description of the incubation setup (line 83-106). We modified the paragraph and add further information to prevent misunderstandings by readers.

"...Soil was immediately sieved (< 2 mm) after sampling and divided into subsamples for further basic soil analytics. Subsample material used for incubation was stored at approximately 8°C....Soil subsamples used for determination of soil basic properties were not used for incubation experiment..."

4. Some minor comments:

Reviewer: L. 164: Do not show any instead "show no"

Response: done

Reviewer: L. 204: What do you mean by "trend" Please, show p-value

Response: We changed "trend" to "tendency". In this case we cited results from another study (de Souza Machado et al. (2018)) to discuss our study results. In our opinion it does not create an added value to cite the results very detailed. Thus, we waived p-values from other studies, because the main focus lies on the confirmation of our results.

Reviewer: PLFA are only biomarkers, not as sensitive as DNA analyses for specific

taxa. The authors should be careful in taking each PLFA biomarker for granted, and they should add a discussion on potential misinterpretations and uncertainties, maybe in an extra paragraph towards the end of the methods section.

Response: We agree with the reviewer and added a paragraph, which shows limitation of the PLFA biomarker approach (line 294-303).

"...Nevertheless, it should be borne in mind that PLFA analyses and laboratory experiments always generate limited results. Fast change of PLFA pattern only allows a determination of actual state of the microbial community structure and it is unreliable to use single PLFA biomarker for taxa detection, which is feasible by deoxyribonucleic acid (DNA) analyses. But compared to gene sequencing or other DNA analyses, PFLA biomarker analysis is rapider and cheaper (Frostegård et al., 2011). Another problem may be the transferability of results generated on laboratory scale under ideal conditions (well-known homogenous plastic fabrics as treatments, simplified and controllable regimes, no rhizosphere, etc.). Also, the single addition of high amounts of microplastics does not reflect the ordinary way how microplastics enter an ecosystem. The accumulation of plastic particles in soils is rather a long and gradual process than a single event, which do not trigger sudden environmental impacts (Rillig et al., 2019)..."

Reviewer: Note that 10Me16:0 is not only used for Actinomycetes, for instance, but has largely been suggested for S utilizing bacteria (see, e.g., work done by R. Evershed and others)

Response: We agree with the reviewer and exclude 10Me16:0 for calculating Actinomycetes. Based on the modification, affected figures (including the statistics) were updated. Please find attached the revised version.

Reviewer: Figure 1 is nice but it does not really relate to the contents of this paper. If the authors want to leave it, I suggest they should go a bit more into detail into the consequences of comparing the different sizes.

Response: In our opinion, figure 1 is an essential feature of the introduction. We mention that no clear definition exists – regarding the size and structure (even in scientific paper) of microplastics (line 65 et seqq.). Therefore, we tried to get a definition for microparticles especially for future studies dealing with the effects of microparticles on soil fauna and flora. Figure 1 serves as a graphical overview about a potential size classification described in different review paper dealing with micro- and nanoplastics. Furthermore, figure 1 displays the potential interaction potentials between soil mineral phase, biosphere and artificial microparticles, which are relevant for our interpretation of the results.

"...The difficulty of highly diverse study structures and test environments due to heterogenic material properties is already reported in related research disciplines like marine and freshwater ecology (Phuong et al., 2016; Rist and Hartmann, 2018). To create a standardize study structure in soil science, we highly recommend for future scientific studies dealing with the effect of artificial microparticles on soil flora and fauna to use the definition and size comparison shown in Fig. 1. Furthermore, a detailed description of microparticle characteristics should be mandatory to show potential interactions between biotic or abiotic soil components and microparticles on different size scales...."

Reviewer: The stirring for microglass and microplastic incorporation into soil likely interfered with soil aggregation? Can it be that this stirring jointly with glass treatment also impaired protozoa? For me this would be a reasonable explanation for the results presented.

Response: All treatments (including the control treatment) were handled exactly the same way to compare effects between different treatments. Effects, caused by handling or laboratory routine, can never be completely eliminated, but due to the analogous sample preparation, potential effects should affect all sample replicates and treatments in a similar way. On the one hand, we are neither able to classify nor prove potential influences caused by handling, but on the other hand the results of the experiment show varying protozoa contents after treatment with different artificial

microparticles (e.g. LD-PE or PS show higher protozoa contents than PA12 or micro-glass). This indicates that it could be possible that stirring inhibit protozoa, but does not explain the question why protozoa are inhibited by microglass. This question still remains open and further research is needed.

References:

Frostegård, Å., Tunlid, A. and Bååth, E.: Use and misuse of PLFA measurements in soils, Soil Biol. Biochem., 43(8), 1621–1625, doi:10.1016/j.soilbio.2010.11.021, 2011.

Phuong, N. N., Zalouk-Vergnoux, A., Poirier, L., Kamari, A., Châtel, A., Mouneyrac, C. and Lagarde, F.: Is there any consistency between the microplastics found in the field and those used in laboratory experiments?, Environ. Pollut., 211, 111–123, doi:10.1016/j.envpol.2015.12.035, 2016.

Rillig, M. C., de Souza Machado, A. A., Lehmann, A. and Klümper, U.: Evolutionary implications of microplastics for soil biota, Environ. Chem., 16(1), 3, doi:10.1071/EN18118, 2019.

Rist, S. and Hartmann, N. B.: Aquatic Ecotoxicity of Microplastics and Nanoplastics: Lessons Learned from Engineered Nanomaterials, in Freshwater Microplastics : Emerging Environmental Contaminants?, edited by M. Wagner and S. Lambert, pp. 25–49, Springer International Publishing, Cham., 2018.

de Souza Machado, A. A., Lau, C. W., Till, J., Kloas, W., Lehmann, A., Becker, R. and Rillig, M. C.: Impacts of Microplastics on the Soil Biophysical Environment, Environ. Sci. Technol., 52(17), 9656–9665, doi:10.1021/acs.est.8b02212, 2018.

[Figure]

[Figure]

**Figure 3.** Phospholipid fatty acids as microbial marker in an incubated Chernozem after 80 days. a) Total bacterial-derived PLFA, b) Total fungal-derived PLFA and c) Sum of total fungal- and bacterial-derived PLFA. Different lowercase letters indicate significant differences between the treatment according to a multiple comparison by the Nemenyi test (n=4, p < 0.05). Please note varying ordinate scales.

**Fig. 1.**

[Figure]

**Figure 4.** Microbial PLFA contents of the individual taxonomic groups of an incubated Chernozem after 80 days. Different lowercase letters indicate significant differences according to a multiple comparison by the Nemenyi test (n=4, p < 0.05). Please note varying ordinate scales.

**Fig. 2.**

---

## Author Comment (AC2) · 28 Feb 2020

Response to comments of anonymous referee #2

Referee comments:

This manuscript investigates the effects of microplastic and microglass particles on the structure of microbial communities in soil using soil microcosms that have been spiked with these contaminants. The issue of micro-particles in the environment is very topical, and while there is a lot of information about the impact of macro-plastics

on wildlife e.g. marine animals, there is relatively little information on the impact of microparticles on microbial populations in terrestrial environments. In this respect, this manuscript is timely. However, the explanation of the experimental design was lacking, and therefore the results should be interpreted with care.

1. Reviewer: Are the authors confident that an incubation period of 80 days was sufficient to observe full effects of the addition of micro-particles?

Response: Bacteria are one of the fast crowing organisms on the world – for instance within a few days, agar plates are fully colonized with bacteria (and fungi). Fungi are, of course a bit slower in its reproduction, but fast enough in order to see an effect after 80 days of incubation. We created an optimal environment for the microorganism (water and temperature conditions). Under natural conditions microorganisms (fast changing wet and dry soil conditions) need a fast reproduction rate in order to survive. Thus, in our opinion 80 days are adequate time to establish a steady microcosm. Although, the microcosm is very artificial (no rhizosphere, macrofauna or variations in temperature or water content).

2. Reviewer: How was 80 days selected as the end point of the experiment? Was it based on published literature or observations?

Response: As in every study, the end point is often set by time and money. In addition, we checked different studies and found, that many of them dealing with soil microorganisms used even much less time.

3. Reviewer: The apparent lack of significant alterations in the bacterial and fungal communities may be due to a relatively short incubation time.

Response: As already explained, microorganisms are fast crowing. Therefore, microbial ecotoxicology test last mostly 7 to 28 days depending on the experimental design. In our opinion, the time period of 80 days is sufficient to establish a stable microcosm and provoke potential treatment effects. But it is conceivable that other microplastic

types (e.g. secondary microplastics) cause stronger impacts on soil microbiology after 80 days as mentioned in line 209-216. In this case, further research is needed.

"...Reasons for missing significant effects between microparticle treatments and the untreated control after 80 days may be found in the conscious choice of primary microplastics, which were not pre-treated to cause a physical degradation (e.g. ultraviolet radiation). Subsequently, microplastics are mostly chemically inert during the experiment due to unaltered chemical and physical properties, which prohibit the exposition of potential ecotoxic components. Nevertheless, the treatment of soil by different microparticles causes changes in microbial communities, albeit not significant. The observed effects are based on complex soil-impurity interactions and studies dealing with the impact of microplastics on soil microbiology are still lacking (Rillig and Bonkowski, 2018; Zhang et al., 2019) and, to our best knowledge, published PLFA or even DNA based studies are still missing..."

4. Reviewer: In addition, the authors did not consider the effects of transfer of the field soil into the lab environment and compartmentalisation of the soils as a cause of the observed changes in PLFAs. This could be remedied if the authors provide PLFA profiles before the soils were used in the microcosms for comparison, or consider such changes in the discussion section.

Response: Potential transfer effects from field to laboratory are not interesting, because all samples are handled the same way und the use of a control version is exactly the reason of your mention and in order to compare effects. Transfer effects can never be completely eliminated, but due to the analogous sample preparation, potential effects should affect all sample replicates in a similar way. Thus, systematic errors are only minor problem in this experimental design.

5. Reviewer: The amounts of microparticles used in the microcosms (1%) is very high compared to what is observed in the field. The authors state that this is comparable to an industrial site, but this is a rare case, and so these results will not be relevant for

most environmental scenarios.

Response: We include line 14-15, 306 and 314 (Abstract, Discussion, Conclusion) that the amounts of microparticles used in our study indicating a worst-case scenario.

6. Reviewer: If the authors think that colonisation on the microplastics could explain the increase in PLFAs, they could use SEM to confirm this, especially when they had already used SEM to characterise the micro-particles at the beginning of the experiment.

Response: Unfortunately, Cryo-SEM is necessary for fungal and bacterial SEM microscopy. This kind of instrumentation is not available in-house and the study was financially limited due to their pioneering character. Thus, we mentioned that our study is a basis for further studies (e. g. line 327-328). Our discussion attempts to explain our observations, but does not prove the assumptions, which is not unusual for experiments dealing with microorganisms.

7. Reviewer: In the discussion section, the authors discuss the changes of PLFAs after the addition of microparticles, but also state that overall, soil organisms were not significantly affected. If the latter is true, then the relative changes are of no consequence. Instead, there should be a discussion on the apparent lack of impact of microplastics on the microbial communities, especially when the literature that they cite points to the contrary. On the other hand, the PLFA may not be able to detect finer microbial community changes that e.g. a DNA-based method will be able to detect – there needs to be a discussion on this. There should also be more of a discussion on why microglass should only affect protozoa and not bacteria. The authors only cite one paper, but it confuses matters as they found that microglass inhibits bacterial growth, which was not the case in the experiment.

Response: We conceive the idea of the reviewer and tried to find studies dealing with effects of natural or artificial particles, which are made of quartz, on protozoa. To our best knowledge no studies were performed in order to investigate this question. We

added a critical evaluation of this result in line 259-261.

"...This harmful effects of microglass particles on protozoa observed in our study are surprisingly, because this indicates that e.g. sand grains in soil, which consist of SiO2, may also have inhibitory effects on protozoa. To our best knowledge no studies were performed in order to investigate this question...."

8. Minor points:

Reviewer: The manuscript could do with a native English speaker to correct the grammar.

Response: done

Reviewer: The paragraph in the discussion section on the effects of micro-particles on macrofauna seems irrelevant when the experiments were about testing microbial populations.

Response: We understand the reviewer's point of view, but our intentions of discussing effects on macrofauna are based on the SEM analyses and show further potential problems caused by microparticles in soil. In the first section of our study we showed the morphology of different microparticles in SEM pictures. We thus included a short discussion on soil fauna due the possible harmful effects on soil fauna originating from those particles.

Reviewer: Figure 1 does not add anything to the manuscript.

Response: In our opinion, figure 1 is an essential feature of the introduction. We mention that no clear definition exists – regarding the size and structure (even in scientific paper) of microplastics (line 65 et seqq.). Therefore, we tried to get a definition for microparticles especially for future studies dealing with the effects of microparticles on soil fauna and flora. Figure 1 serves as a graphical overview about a potential size classification described in different review paper dealing with micro- and nanoplastics. Furthermore, figure 1 displays the potential interaction potentials between soil mineral

phase, biosphere and artificial microparticles, which are relevant for our interpretation of the results.

"...The difficulty of highly diverse study structures and test environments due to heterogenic material properties is already reported in related research disciplines like marine and freshwater ecology (Phuong et al., 2016; Rist and Hartmann, 2018). To create a standardize study structure in soil science, we highly recommend for future scientific studies dealing with the effect of artificial microparticles on soil flora and fauna to use the definition and size comparison shown in Fig. 1. Furthermore, a detailed description of microparticle characteristics should be mandatory to show potential interactions between biotic or abiotic soil components and microparticles on different size scales...."

Reviewer: I don't understand the use of lowercase a and b to denote p-values. Better to state the p-values.

Response: In figures showing graphs it is an adequate way to use letters (or other symbols) to indicate homogeneous subset, which were defined by using a multiple comparison between the different treatment level (Post-Hoc Test). Using p-values instead of homogenous subsets would require tables instead of box-plot graphs, which we do not prefer due to (in our opinion) a better visibility of several statistical parameters. We add detailed information in line 162-163 to enhance the comprehension.

"...Residuals of each linear model were checked graphically for homoscedasticity and normal distribution to validate the model performance. Because of widespread heteroscedasticity and bad model performances, differences in PLFA marker contents between treatments of each taxonomic microbial group were statistically evaluated using the Kruskal-Wallis rank sum test. Nemenyi test was performed for multiple comparison between the treatment levels in case of a significant ($p \leq 0.05$) treatment effect in the Kruskal-Wallis test. Different lowercase letters were used to illustrate significant differences between homogeneous subsets...."

Reviewer: The use of the plastic cylinders to adjust water holding capacity will also

contaminate the soils with plastic.

Response: Subsamples were used for detection of water holding capacity, which were not used for the incubation. Thus, a risk of contamination with microplastic can be excluded. We add further information to prevent misunderstandings by readers (line 118-119).

". . .Soil subsamples used for determination of soil basic properties were not used for incubation experiment. . ."

Reviewer: 'WHC' should be defined.

Response: The analytical approach is described in line 107-110. In our opinion, the target group of this journal have professional expertise in soil science. Thus, function of water holding capacity in soil is generally known.

References:

Phuong, N. N., Zalouk-Vergnoux, A., Poirier, L., Kamari, A., Châtel, A., Mouneyrac, C. and Lagarde, F.: Is there any consistency between the microplastics found in the field and those used in laboratory experiments?, Environ. Pollut., 211, 111–123, doi:10.1016/j.envpol.2015.12.035, 2016.

Rillig, M. C. and Bonkowski, M.: Microplastic and soil protists: A call for research, Environ. Pollut., 241, 1128–1131, doi:10.1016/j.envpol.2018.04.147, 2018.

Rist, S. and Hartmann, N. B.: Aquatic Ecotoxicity of Microplastics and Nanoplastics: Lessons Learned from Engineered Nanomaterials, in Freshwater Microplastics : Emerging Environmental Contaminants?, edited by M. Wagner and S. Lambert, pp. 25–49, Springer International Publishing, Cham., 2018.

Zhang, S., Wang, J., Liu, X., Qu, F., Wang, X., Wang, X., Li, Y. and Sun, Y.: Microplastics in the environment: A review of analytical methods, distribution, and biological effects, TrAC Trends Anal. Chem., 111, 62–72, doi:10.1016/j.trac.2018.12.002, 2019.

---

## Author Response (AR1)

**Authors response to referee and editor comments on the manuscript: Effects of microplastic and microglass particles on soil microbial community structure in an arable soil (Chernozem)**

**Response to comments of anonymous referee #1**

Referee comments:

Increasing loads of microplastic waste potentially burden our soils. In this regard the paper is timely, as it investigates potential effects of microplastic and microglass pollution on soil microbial community in a laboratory incubation study. The manuscript is concise, very well written and organized, and it has improved in regard to a previous version. However, still the paper includes the risk of presenting artificial results, which should be very openly discussed.

The shortcomings refer to:

1. Microplastic loads: The authors state that they refer to microplastic loads near industrial areas. However, 12 t ha$^{-1}$ is a huge amount, far from being realistic. The authors should spell out clearly, also in abstract and conclusions, that their data refer to worst-case conditions that do not necessarily apply to common plastic and microglass loads in Discussion paper agricultural soils, because concentrations exceed natural loads at least by a factor of about 10.000!

   **Response:** We include line 14-15, 92-93, 294-296 and 302-303 (Abstract, Discussion, Conclusion) that the amounts of microparticles used in our study indicating a worst-case scenario.

2. I like the finding for protozoa, and appreciate that an explanation is offered related to the hydrophilic surface. Nevertheless, why should this apply to glass but not to increased amounts of sand grains? Can enhanced amoutns of quartz grains also be toxic for protozoa and has this been published before? And if not, why should the glass be more toxic than pure sand? Here the authors should elaborate the physiological explanations a bit more in detail and also outline why microglass should be toxic whereas quartz particles in the fine sand fraction is apparently not (or is it?). It is also not clear why a specific toxicity should only apply for protozoa while one of their main food sources, bacteria, are not affected.

   **Response:** We conceive the idea of the reviewer and tried to find studies dealing with effects of natural or artificial particles, which are made of quartz, on protozoa. To our best knowledge no studies were performed in order to investigate this question. We added a critical evaluation of this result in line 269-279.

   *"…Nevertheless, these harmful effects of microglass particles on protozoa observed in our study are surprisingly, because this indicates that e.g. sand grains in soil, which consist of $SiO_2$, may also have inhibitory effects on protozoa. To our best knowledge no studies were performed in order to investigate this question…."*

3. Experiment conditions: Usually soil has to be stored cool but should not be air-dried. Air-drying soil prior to incubation in known that it includes the risks of artifacts, even if pre-incubated. The authors should discuss this issue based on some literature which investigated related effects of sieving and air-drying for a range of microbial parameters

**Response:** The reviewer is right. However, the soil samples in our experiment were not air-dried prior to incubation. In our opinion, the reviewer misunderstood the description of the incubation setup (line 83-106). We modified the paragraph and add further information to prevent misunderstandings by readers.

4. Some minor comments:

- L. 164: Do not show any instead "show no"

**Response:** done

- L. 204: What do you mean by "trend" Please, show p-value

**Response:** We changed "trend" to "tendency". In this case we cited results from another study (de Souza Machado et al. (2018)) to discuss our study results. In our opinion it does not create an added value to cite the results very detailed. Thus, we waived p-values from other studies, because the main focus lies on the confirmation of our results.

- PLFA are only biomarkers, not as sensitive as DNA analyses for specific taxa. The authors should be careful in taking each PLFA biomarker for granted, and they should add a discussion on potential misinterpretations and uncertainties, maybe in an extra paragraph towards the end of the methods section.

**Response:** We agree with the reviewer and added a paragraph, which shows limitation of the PLFA biomarker approach (line 283-292).

*"…Nevertheless, it should be borne in mind that PLFA analyses and laboratory experiments always generate limited results. Fast change of PLFA pattern only allows a determination of actual state of the microbial community structure and it is unreliable to use single PLFA biomarker for taxa detection, which is feasible by deoxyribonucleic acid (DNA) analyses. But compared to gene sequencing or other DNA analyses, PFLA biomarker analysis is rapider and cheaper (Frostegård et al., 2011). Another problem may be the transferability of results generated on laboratory scale under ideal conditions (well-known homogenous plastic fabrics as treatments, simplified and controllable regimes, no rhizosphere, etc.). Also, the single addition of high amounts of microplastics does not reflect the ordinary way how microplastics enter an ecosystem. The accumulation of plastic particles in soils is rather a long and gradual process than a single event, which do not trigger sudden environmental impacts (Rillig et al., 2019)…"*

- Note that 10Me16:0 is not only used for Actinomycetes, for instance, but has largely been suggested for S utilizing bacteria (see, e.g., work done by R. Evershed and others)

**Response:** We agree with the reviewer and exclude 10Me16:0 for calculating Actinomycetes. Based on the modification, affected figures (including the statistics) were updated.

- Figure 1 is nice but it does not really relate to the contents of this paper. If the authors want to leave it, I suggest they should go a bit more into detail into the consequences of comparing the different sizes.

**Response:** In our opinion, figure 1 is an essential feature of the introduction. We mention that no clear definition exists – regarding the size and structure (even in scientific paper) of microplastics (line 65 et seqq.). Therefore, we tried to get a definition for microparticles especially for future studies dealing with the effects of microparticles on soil fauna and flora. Figure 1 serves as a graphical overview about a potential size classification described in different review paper dealing with micro- and nanoplastics. Furthermore, figure 1 displays the potential interaction potentials between soil mineral phase, biosphere and artificial microparticles, which are relevant for our interpretation of the results.

*"…The difficulty of highly diverse study structures and test environments due to heterogenic material properties is already reported in related research disciplines like marine and freshwater ecology (Phuong et al., 2016; Rist and Hartmann, 2018). To create a standardize study structure in soil science, we highly recommend for future scientific studies dealing with the effect of artificial microparticles on soil flora and fauna to use the definition and size comparison shown in Fig. 1. Furthermore, a detailed description of microparticle characteristics should be mandatory to show potential interactions between biotic or abiotic soil components and microparticles on different size scales…."*

- The stirring for microglass and microplastic incorporation into soil likely interfered with soil aggregation? Can it be that this stirring jointly with glass treatment also impaired protozoa? For me this would be a reasonable explanation for the results presented.

**Response:** All treatments (including the control treatment) were handled exactly the same way to compare effects between different treatments. Effects, caused by handling or laboratory routine, can never be completely eliminated, but due to the analogous sample preparation, potential effects should affect all sample replicates and treatments in a similar way. On the one hand, we are neither able to classify nor prove potential influences caused by handling, but on the other hand the results of the experiment show varying protozoa contents after treatment with different artificial microparticles (e.g. LD-PE or PS show higher protozoa contents than PA12 or microglass). This indicates that it could be possible that stirring inhibit protozoa, but does not explain the question why protozoa are inhibited by microglass. This question still remains open and further research is needed.

**Response to comments of anonymous referee #2**

Referee comments:

This manuscript investigates the effects of microplastic and microglass particles on the structure of microbial communities in soil using soil microcosms that have been spiked with these contaminants. The issue of micro-particles in the environment is very topical, and while there is a lot of information about the impact of macro-plastics on wildlife e.g. marine animals, there is relatively little information on the impact of microparticles on microbial populations in terrestrial environments. In this respect, this manuscript is timely. However, the explanation of the experimental design was lacking, and therefore the results should be interpreted with care.

1. Are the authors confident that an incubation period of 80 days was sufficient to observe full effects of the addition of micro-particles?

   **Response:** Bacteria are one of the fast crowing organisms on the world – for instance within a few days, agar plates are fully colonized with bacteria (and fungi). Fungi are, of course a bit slower in its reproduction, but fast enough in order to see an effect after 80 days of incubation. We created an optimal environment for the microorganism (water and temperature conditions). Under natural conditions microorganisms (fast changing wet and dry soil conditions) need a fast reproduction rate in order to survive. Thus, in our opinion 80 days are adequate time to establish a steady microcosm. Although, the microcosm is very artificial (no rhizosphere, macrofauna or variations in temperature or water content).

2. How was 80 days selected as the end point of the experiment? Was it based on published literature or observations?

   **Response:** As in every study, the end point is often set by time and money. In addition, we checked different studies and found, that many of them dealing with soil microorganisms used even much less time.

3. The apparent lack of significant alterations in the bacterial and fungal communities may be due to a relatively short incubation time.

   **Response:** As already explained, microorganisms are fast crowing. Therefore, microbial ecotoxicology test last mostly 7 to 28 days depending on the experimental design. In our opinion, the time period of 80 days is sufficient to establish a stable microcosm and provoke potential treatment effects. But it is conceivable that other microplastic types (e.g. secondary microplastics) cause stronger impacts on soil microbiology after 80 days as mentioned in line 209-216. In this case, further research is needed.

   *"…Reasons for missing significant effects between microparticle treatments and the untreated control after 80 days may be found in the conscious choice of primary microplastics, which were not pre-treated to cause a physical degradation (e.g. ultraviolet radiation). Subsequently, microplastics are mostly chemically inert during the experiment due to unaltered chemical and physical properties, which prohibit the exposition of potential ecotoxic components. Nevertheless, the treatment of soil by different microparticles causes changes in microbial communities, albeit not significant. The observed effects are based on complex soil-impurity interactions and studies dealing with the impact of microplastics on soil*

*microbiology are still lacking* (Rillig and Bonkowski, 2018; Zhang et al., 2019) *and, to our best knowledge, published PLFA or even DNA based studies are still missing…"*

4. In addition, the authors did not consider the effects of transfer of the field soil into the lab environment and compartmentalisation of the soils as a cause of the observed changes in PLFAs. This could be remedied if the authors provide PLFA profiles before the soils were used in the microcosms for comparison, or consider such changes in the discussion section.

   **Response:** Potential transfer effects from field to laboratory are not interesting, because all samples are handled the same way und the use of a control version is exactly the reason of your mention and in order to compare effects. Transfer effects can never be completely eliminated, but due to the analogous sample preparation, potential effects should affect all sample replicates in a similar way. Thus, systematic errors are only minor problem in this experimental design.

5. The amounts of microparticles used in the microcosms (1%) is very high compared to what is observed in the field. The authors state that this is comparable to an industrial site, but this is a rare case, and so these results will not be relevant for most environmental scenarios.

   **Response:** We include line 14-15, 92-93, 294-296 and 302-303 (Abstract, Discussion, Conclusion) that the amounts of microparticles used in our study indicating a worst-case scenario.

6. If the authors think that colonisation on the microplastics could explain the increase in PLFAs, they could use SEM to confirm this, especially when they had already used SEM to characterise the micro-particles at the beginning of the experiment.

   **Response:** Unfortunately, Cryo-SEM is necessary for fungal and bacterial SEM microscopy. This kind of instrumentation is not available in-house and the study was financially limited due to their pioneering character. Thus, we mentioned that our study is a basis for further studies (e. g. line 327-328). Our discussion attempts to explain our observations, but does not prove the assumptions, which is not unusual for experiments dealing with microorganisms.

7. In the discussion section, the authors discuss the changes of PLFAs after the addition of microparticles, but also state that overall, soil organisms were not significantly affected. If the latter is true, then the relative changes are of no consequence. Instead, there should be a discussion on the apparent lack of impact of microplastics on the microbial communities, especially when the literature that they cite points to the contrary. On the other hand, the PLFA may not be able to detect finer microbial community changes that e.g. a DNA-based method will be able to detect – there needs to be a discussion on this. There should also be more of a discussion on why microglass should only affect protozoa and not bacteria. The authors only cite one paper, but it confuses matters as they found that microglass inhibits bacterial growth, which was not the case in the experiment.

   **Response:** We conceive the idea of the reviewer and tried to find studies dealing with effects of natural or artificial particles, which are made of quartz, on protozoa. To our best knowledge no studies were performed in order to investigate this question. We added a critical evaluation of this result in line 269-271.

*"…This harmful effects of microglass particles on protozoa observed in our study are surprisingly, because this indicates that e.g. sand grains in soil, which consist of $SiO_2$, may also have inhibitory effects on protozoa. To our best knowledge no studies were performed in order to investigate this question…."*

8. Minor points:

The manuscript could do with a native English speaker to correct the grammar.

**Response:** done

The paragraph in the discussion section on the effects of micro-particles on macrofauna seems irrelevant when the experiments were about testing microbial populations.

**Response:** We understand the reviewer's point of view, but our intentions of discussing effects on macrofauna are based on the SEM analyses and show further potential problems caused by microparticles in soil. In the first section of our study we showed the morphology of different microparticles in SEM pictures. We thus included a short discussion on soil fauna due the possible harmful effects on soil fauna originating from those particles.

Figure 1 does not add anything to the manuscript.

**Response:** In our opinion, figure 1 is an essential feature of the introduction. We mention that no clear definition exists – regarding the size and structure (even in scientific paper) of microplastics (line 65 et seqq.). Therefore, we tried to get a definition for microparticles especially for future studies dealing with the effects of microparticles on soil fauna and flora. Figure 1 serves as a graphical overview about a potential size classification described in different review paper dealing with micro- and nanoplastics. Furthermore, figure 1 displays the potential interaction potentials between soil mineral phase, biosphere and artificial microparticles, which are relevant for our interpretation of the results.

**"…***The difficulty of highly diverse study structures and test environments due to heterogenic material properties is already reported in related research disciplines like marine and freshwater ecology (Phuong et al., 2016; Rist and Hartmann, 2018). To create a standardize study structure in soil science, we highly recommend for future scientific studies dealing with the effect of artificial microparticles on soil flora and fauna to use the definition and size comparison shown in Fig. 1. Furthermore, a detailed description of microparticle characteristics should be mandatory to show potential interactions between biotic or abiotic soil components and microparticles on different size scales…."*

I don't understand the use of lowercase a and b to denote p-values. Better to state the p-values.

**Response:** In figures showing graphs it is normal and an adequate way to use letters (or other symbols) to indicate homogeneous subset, which were defined using a multiple comparison between the different treatment level (Post-Hoc Test). Using p-values instead of homogenous subsets would require tables instead of box-plot graphs, which we do not prefer due to (in our opinion) a better visibility of several statistical parameters. We add detailed information in line 160-163 to enhance the comprehension.

*"…Residuals of each linear model were checked graphically for homoscedasticity and normal distribution to validate the model performance. Because of widespread heteroscedasticity and bad model performances, differences in PLFA marker contents between treatments of each taxonomic microbial group were statistically evaluated using the Kruskal-Wallis rank sum test. Dunn's test was performed for multiple comparison between the treatment levels in case of a significant ($p \leq 0.05$) treatment effect in the Kruskal-Wallis test* (Dunn, 1964) *Holm method was used to control the family-wise-error rate caused by the pairwise multiple comparisons* (Holm, 1979)*. Different lowercase letters were used to illustrate significant differences between homogeneous subsets. …."*

The use of the plastic cylinders to adjust water holding capacity will also contaminate the soils with plastic.

**Response:** Subsamples were used for detection of water holding capacity, which were not used for the incubation. Thus, a risk of contamination with microplastic can be excluded. We add further information to prevent misunderstandings by readers (line 117-118).

*"…Soil subsamples used for determination of soil basic properties were not used for incubation experiment…"*

'WHC' should be defined.

**Response:** The analytical approach is described in line 116-118. In our opinion, the target group of this journal have professional expertise in soil science. Thus, function of water holding capacity in soil is generally known.

**Response to comments of the editor**

Editor comments:

1. Reviewer 2, point 7 needs addressing more thoroughly. The author response only deals with the 2nd part of the point on microglass. The first point the reviewer is making is that if you report non-significant effects of microplastics on PLFAs, then even if there are trends in the data, the discussion should focus on why there are no significant effects. This is especially true given the very high input rates of microplastics which have been used.

   **Response:** A paragraph containing a probable explanation for non-significant effects of microplastics on PLFAs was added (Line 209-214).

   "…Reasons for missing significant effects between microparticle treatments and the untreated control after 80 days may be found in the conscious choice of primary microplastics, which were not pre-treated to cause a physical degradation (e.g. ultraviolet radiation). Subsequently, microplastics are mostly chemically inert during the experiment due to unaltered chemical and physical properties, which e. g. prohibit the exposition of potential ecotoxic compounds. Nevertheless, the treatment of soil by different microparticles causes changes in microbial communities, albeit not significant…"

2. Reviewer 2, point 8 states: The paragraph in the discussion section on the effects of micro-particles on macrofauna seems irrelevant when the experiments were about testing microbial populations. Editor – I would agree with the reviewer that this paragraph should be removed or reduced to a single statement as it is not the focus of the paper.

   **Response:** The paragraph was removed.

3. Both reviewers indicate that figure 1 is unnecessary and should be removed. The figure contains information which can be gained from the references cited, and also the figure is not used as a conceptual framework for the experiments or referred to elsewhere in the paper, beyond one mention in the introduction. It should therefore be removed from the manuscript.

   **Response:** Figure 1 was removed and all paragraphs referring to the figure were modified or removed.

4. Regarding the attribution of specific PLFA indicators for general fungi and arbuscular mycorrhizal fungi these attributions are incorrect (see Frostegard et al 2011 for discussion and correct use). The authors use General fungi: 18:2ω6,9, 18:1ω9c, 20:1ω9c; and arbuscular mycorrhizal fungi (AMF): 16:1ω5c. For general fungi I have never seen 20:1ω9c ascribed to fungi so would question if this is accurate (Frostegard et al 2011). For arbuscular fungi it is widely published that PLFA 16:1ω5c can only be used in for AMF in non-soil systems as it also present in bacteria (Olsson et al 1999). It is the NLFA which should be used in soils (Frostegard et al 2011, Soil Biol Biochem). These attributions need correcting or strongly justifying.

   **Response:** Editor recommendations regarding to the attribution of specific PLFA indicators were implemented (Line 139-146). 20:1ω9c as fungal biomarker is no longer been used. 16:1ω5c as AMF

biomarker is now classed as marker for total PLFA due to discrepancy of the origin in soil systems (Frostegård et al., 2011; Olsson, 1999). 10Me16:0 was also classed as marker for total PLFA (reviewer 1 commented that 10Me16:0 is not only specific for ACT but also for S utilizing bacteria). Based on the modifications, affected figures (including the statistics) were updated.

[revised manuscript text omitted]